# The Unique Roles of Ion Channels in Pluripotent Stem Cells in Response to Biological Stimuli

**DOI:** 10.3390/biology13121043

**Published:** 2024-12-13

**Authors:** Taku Kaitsuka

**Affiliations:** School of Pharmacy at Fukuoka, International University of Health and Welfare, Enokizu 137-1, Okawa 831-8501, Fukuoka, Japan; kaitsuka@iuhw.ac.jp

**Keywords:** pluripotent stem cells, embryonic stem cells, induced pluripotent stem cells, ion channels, pluripotency, self-renewal, proliferation

## Abstract

Pluripotent stem cells like embryonic stem and induced pluripotent stem cells have unique properties of pluripotency and self-renewal. The major role of membrane-bound receptors and their downstream signaling after binding their ligands (leukemia inhibitory factor and growth factors) in maintaining their properties is well documented. So far, the roles of ion and ion channels in pluripotent stem cells have not been much focused on or investigated. Cellular ion homeostasis is essential for all types of cells, and Ca^2+^ and Mg^2+^-dependent signaling is also important for cellular events such as metabolism, stress response, cell cycle and survival. In this review, presently restricted insights about ion channels are introduced, and other candidates for ion channels that could regulate pluripotency and self-renewal are extracted for further understanding of pluripotent stem cells.

## 1. Introduction

Pluripotent stem cells (PSCs) are quite unique cells that have self-renewal and pluripotency abilities. A significant molecular event for pluripotency is transcriptional regulation mediated by pluripotency factors, OCT4 and NANOG. Epigenetic regulation of chromatins in pluripotency and differentiation genes is also an essential factor for the abilities of PSCs. Negative regulation of differentiation genes is mediated by histone H3 lysine 27 methylation by polycomb groups. Those main events generally occur in the nucleus of cells.

Fundamentally, environmental signals from culture medium to PSCs regulate their pluripotency. In naive PSCs of mouse embryonic stem cells (ESCs), the leukemia inhibitory factor (LIF)-JAK-STAT pathway regulates their self-renewal and pluripotency [1]. LIF binds to the LIF receptor, which forms a complex with the receptor glycoprotein 130 [2]. In primed PSCs of human ESCs, the transforming growth factor-β (TGF-β)/Activin A/Nodal pathway regulates their pluripotency by activating Smad2/3 [3]. In vitro, the basic fibroblast growth factor (bFGF) is essential for human PSC stemness and self-renewal maintenance [4,5,6]. bFGF activates the phosphatidylinositol 3-kinase (PI3K)/AKT pathway, inhibiting ERK activity, WNT signaling and dephosphorylation of glycogen synthase kinase-3β (GSK-3β) [7,8]. Other signals stimulating membrane proteins are involved in the unique function of PSCs, mainly proliferation ability and regulation of the pluripotent factors. For example, disruption of transient receptor potential channel 7 (TRPM7) causes proliferation arrest in mouse embryonic stem cells [9]. The voltage-gated potassium channel (Kv) and membrane potential affect self-renewal in mouse ESCs, and the Kv blocker attenuates its cell cycle [10].

To date, there are few reviews in which the overall argument about the functions and roles of ion channels in PSCs are included. In this review, insights about the functions of ion channels in PSCs are summarized, and their roles are discussed. Additionally, the candidate channels that regulate the property of PSCs are extracted from previous reports using undifferentiated and differentiated cells.

## 2. Ion Channels

Ion channels are proteins that facilitate the movement of ions across cell membranes, playing a crucial role in cellular functions. These channels are found in both the plasma membrane and intracellular membranes and are expressed in organisms ranging from bacteria to humans, where they are essential for life. In general, excitable cells, including neurons, muscle cells and touch receptor cells, express these channels and use them to generate electrical signals. In non-excitable cells, ion channels regulate intracellular ion levels and affect the activities of intracellular proteins and their functions. Opening of ion channels is mediated by the stimulation from external or internal ligands such as second messengers, voltage, temperature and mechanical stresses.

Cellular ions include sodium (Na^+^), potassium (K^+^), calcium (Ca^2+^), magnesium (Mg^2+^) and chloride (Cl^−^). Basically, Na^+^ and K^+^ regulate the membrane potential of the plasma membrane. Ca^2+^ acts as a second messenger and activates Ca^2+^-dependent enzymes. Mg^2+^ is essential for Mg^2+^-dependent molecules and proteins like adenosine triphosphate (ATP) and Na-K-ATPase. Cl^−^ is the main anion outside the cell and maintains osmotic pressure similar to the above ions.

## 3. Ion Channels Expressed in Pluripotent Stem Cells

ESCs express a small part of ion channels compared to tissue stem cells and somatic excitable/non-excitable cells [11]. Heatmaps in Figure 1, Figure 2, Figure 3, Figure 4 and Figure 5 show the gene expression levels between human ESCs and their differentiated embryoid bodies (EBs) from the dataset GDS5408 of the National Center for Biotechnology Information (NCBI) website [12]. In this dataset, pluripotency gene *NANOG* in human ESCs is confirmed to be higher than their EBs, while the differentiation gene *SOX17* (endoderm), *MESP1* (mesoderm) and *SOX1* (ectoderm) are higher in EBs than in ESCs (Figure 1). In the case of ion channels, levels of almost all channel genes are higher in EBs than in ESCs, while some of them become lower after differentiation to EBs (Figure 2, Figure 3, Figure 4 and Figure 5). In this review, we focus on the genes in both cell lines (H1 and H9) of ESCs, which decrease less than 0.8-fold compared to their EBs. Specifically, the expression levels of *CLIC4* and *CLIC5* decreased in differentiated EBs of both cell lines (Figure 2). Each gene encodes chloride intracellular channel proteins 4 and 5, respectively. *HCN4*, encoding the hyperpolarization-activated cyclic nucleotide-gated potassium channel 4, also decreased in EBs (Figure 2). Among potassium channels, 19 genes decreased in EBs (Figure 3). These genes and corresponding product proteins are *KCNE3* (voltage-gated potassium channel MiRP2), *KCNH1* (Kv10.1), *KCNH2* (Kv11.1), *KCNH3* (Kv12.2), *KCNH6* (Kv11.2), *KCNJ10* (ATP-sensitive inwardly rectifying potassium channel Kri4.1), *KCNJ11* (Kir6.2), *KCNK2* (two-pore domain potassium channel TREK-1), *KCNK4* (TWIK-related arachidonic acid-stimulated potassium channel TRAAK), *KCNK5* (TWIK-related acid-sensitive potassium channel TASK-2), *KCNK6* (inward rectifying potassium channel protein TWIK-2), *KCNK7* (double-pore potassium channel DP3), *KCNK12* (tandem pore domain halothane-inhibited potassium channel THIK-2), *KCNK17* (TWIK-related acid-sensitive potassium channel TASK-4), *KCNMA1* (calcium-activated potassium channel Kca1.1), *KCNQ1* (voltage-gated potassium channel Kv7.1), *KCNQ2* (Kv7.2), *KCNS1* (Kv9.1) and *KCNS3* (Kv9.3). Among sodium channels, *SCN4A*, *SCN5A* and *SCN8A*, which code voltage-gated sodium channels Nav1.4, Nav1.5 and Nav1.6, are decreased genes in EBs (Figure 4B). *TMC8*, encoding transmembrane channels like 8 and transient receptor potential channels *TRPM4*, *TRPML2* and *TRPV3* also decreased in EBs (Figure 4C and Figure 5). Based on the data on changes in expression levels during differentiation, those channels are suggested to have some roles in the unique function of human ESCs. Supporting this speculation, Zhang et al. reviewed that many types of potassium channels are detected and function in stem cells, including human ESCs [11].

There are several studies about the existence of membrane proteins, including ion channels in human and mouse PSCs [14]. Previously, Jiang et al. analyzed the functional importance of ion channels in human and mouse ESCs [15]. In this report, microarray analysis was performed on human-induced pluripotent stem cells (iPSCs), ESCs and MSCs. In iPSCs, *CACNA1H*, *CACNA2D1*, *CACNA2D2*, *CACNG4*, *CACNG7* coding voltage-gated calcium channel Cav3.2, Cavα-2/δ-1, Cavα-2/δ-2, TARPγ-4, TARPγ-7, *SCN8A* coding voltage-gated sodium channel Nav1.6, *KCNC4*, *KCNK1*, *KCNK12*, *KCNK5*, *KCNK6*, *KCNQ2*, *KCNS3*, *KCNMB4*, *KCNN2* coding voltage-gated potassium channel Kv3.4, inward rectifying potassium channel TWIK-1, tandem pore domain halothane-inhibited potassium channel THIK-2, TASK-2, TWIK-2, Kv7.2, Kv9.3, BKβ4, small conductance calcium-activated potassium channel SK2 and *HCN4* were significantly expressed. The ion channel transcriptome of iPSCs could not be distinguished from that of ESC lines. In Figure 4B, *SCN8A* is detected as it decreased during differentiation to EBs. Similarly, *KCNK5*, *KCNK6*, *KCNK12*, *KCNQ2* and *KCNS3* gens decreased in EBs (Figure 3A,B).

On the other hand, studies on specific channels in human PSCs have been performed, as described in the following sections.

## 4. Sodium Channels

Voltage-gated sodium channel subunits are encoded by *SCN1A* (Nav1.1), *SCN2A* (Nav1.2), *SCN3A* (Nav1.3), *SCN4A* (Nav1.4), *SCN5A* (Nav1.5), *SCN8A* (Nav1.6), *SCN9A* (Nav1.7) and *SCN10A* (Nav1.8). The properties and importance of these subunits are well-reviewed by Meisler et al. [16]. Generally, the knockout of a single gene does not cause embryonic lethality [16], showing that other genes compensate for each other in embryonic development. Based on composed data from microarray analysis [12] and previous reports about the expression of ion channels [15], *SCN8A* seems to have the main role in the transmission of Na^2+^ in PSCs.

### The Role of Sodium Channels in Pluripotent Stem Cells

As mentioned above, Jiang et al. performed transcriptomic analysis of ion channel genes in human ESCs, iPSCs and mesenchymal stem cells (MSCs) [15]. A total of 158 voltage-dependent ion channels were found to be expressed in iPSCs. Among them, only *SCN8A* was a sodium channel detected in human iPSCs by RT-PCR analysis. However, in the experiment with patch-clamp analysis, the current of the voltage-gated sodium channel could not be detected in human ESCs or iPSCs [15,17]. It is assumed that voltage-gated sodium channels, including *SCN8A*, are not essential in maintaining the properties of human PSCs.

## 5. Potassium Channels

Potassium channels consist of numerous families and genes as voltage-gated (*KCNA*, *KCNB*, *KCNC*, *KCND*, *KCNE*, *KCNF*, *KCNG*, *KCNH*, *KCNI*, *KCNQ*, *KCNS* and *KCNV*), inwardly rectifying (*KCNJ*), two-pore domain (*KCNK*), calcium-activated (*KCNM*, *KCNN*, *KCNU*) and sodium-activated (*KCNT*) types. The structure and function of potassium channels have been well-reviewed in several papers [18,19]. Potassium channels mainly function in excitable cells like neurons and cardiomyocytes, and their dysfunction causes psychiatric disorders and cardiovascular diseases in humans. In PSCs, many genes of *KCNE3*, *KCNH1*, *KCNH2*, *KCNH3*, *KCNH6*, *KCNJ10*, *KCNJ11*, *KCNK2*, *KCNK4*, *KCNK5*, *KCNK6*, *KCNK7*, *KCNK12*, *KCNK17*, *KCNMA1*, *KCNQ1*, *KCNQ2*, *KCNS1* and *KCNS3* seem to exist in the pluripotent state of human ESCs via microarray analysis and have some physiological roles (Figure 3).

### The Role of Potassium Channels in Pluripotent Stem Cells

Jiang et al. found that *KCNC4*, *KCNK1*, *KCNK12*, *KCNK5*, *KCNK6*, *KCNQ2*, *KCNS3*, *KCNMB4* and *KCNN2* genes are expressed in human iPSCs by microarray analysis, and these are confirmed by RT-PCR, except *KCNK12* [15]. Non-selective potassium channel blocker tetraethylammonium (TEA)-sensitive delayed rectifier K^+^ currents (I_KDR_) are observed in human ESCs and iPSCs, whereas voltage-gated sodium channel and voltage-gated calcium channel currents cannot be measured in both ESCs or iPSCs [15,17]. As a contribution of potassium channel to PSCs properties, it is shown that TEA and a known blocker of several potassium channel subtypes, 4-aminopyridine, inhibits human ESCs and iPSCs proliferation [15,17]. From this work, potassium channels seem to have an important role in human PSCs.

Recently, Sempou et al. showed that membrane voltage (Vm) via a member of potassium voltage-gated channel Kv11.2 (also called human ether-a-go-go 2 protein) encoded by *KCNH6* is essential for the exit from pluripotency [20]. Membrane depolarization causes elevation of intracellular Ca^2+^ via the opening of voltage-gated calcium channels and maintains the pluripotency in Xenopus oocytes and human ESCs. Thus, the depolarization of Vm via inhibition of Kv11.2 leads to the loss of ectodermal and mesodermal cell fates due to the persistence of pluripotency with elevated expression of pluripotency genes. In this depolarization-induced effect, the mTOR pathway is found to be downstream signaling. Namely, the polarization of membrane potential via the Kv11.2 channel promotes the exit from pluripotency and the activation of differentiated cell fates.

## 6. Calcium Channels

Voltage-gated calcium channel subunits are encoded by *CACNA1A* (Cav2.1), *CACNA1C* (Cav1.2), *CACNA1H* (Cav3.2), *CACNA2D1* (Cavα-2/δ-1) and *CACNB3* (Cavβ-3). The importance of calcium channels and Ca^2+^ signaling in ESCs is reviewed in some papers [21,22,23,24].

### The Role of Calcium Channels in Pluripotent Stem Cells

In previous reports, voltage-gated calcium channel currents are not detected in human PSCs [15,17]. However, Ca^2+^ signaling is shown to be important for the properties of PSCs. Specifically, Ca^2+^ is an important second messenger involved in the maintenance of the pluripotency and self-renewal of human ESCs, as shown by several reports [21,25,26], while these are mainly mediated by store-operated Ca^2+^ entry via store-operated calcium channels as TRPC and ORAI channels [25].

## 7. Transient Receptor Potential Channels

In mammals, the TRP channel family is composed of transient receptor potential ankyrin (TRPA), canonical (TRPC), melastatin (TRPM), mucolipin (TRPML), polycystin (TRPP) and vanilloid (TRPV) channels encoded by *TRPA1*, *TRPC1*, *TRPC2*, *TRPC3*, *TRPC4*, *TRPC5*, *TRPC6*, *TRPC7*, *TRPM1*, *TRPM2*, *TRPM3*, *TRPM4*, *TRPM5*, *TRPM6*, *TRPM7*, *TRPM8*, *TRPML1*, *TRPML2*, *TRPML3*, *TRPP1* (*PKD2*), *TRPP2* (*PKD2L1*), *TRPP3* (*PKD2L2*), *TRPV1*, *TRPV2*, *TRPV3*, *TRPV4*, *TRPV5* and *TRPV6*. The basic and latest knowledge about TRP channels is reviewed by Cox et al. [27].

### 7.1. The Role of TRPC Channels in Pluripotent Stem Cells

TRPC3 knockout (KO) reduces the level of pluripotency marker Oct4 in mouse ESCs [28]. Also, TRPC3 KO induces apoptosis and the disruption of the mitochondrial membrane potential in an undifferentiated state and inhibits neural differentiation. TRPC3 is a non-selective cation channel which is permeable to both Na^+^ and Ca^2+^. Therefore, Hao et al. suggested that TRPC3 activity might be required for the survival and maintenance of pluripotency in mouse ESCs via the modulation of cellular Ca^2+^ [28].

### 7.2. The Role of TRPM Channels in Pluripotent Stem Cells

Among TRPM channels, TRPM4 and TRPM8 KO cell lines in human iPSCs or ESCs exhibit normal morphology, pluripotency and karyotype, showing both channels are not essential for general properties of human PSCs [29,30]. Based on previous reports, TRPM7 seems to be an important TRPM channel in PSCs. TRPM7 is a unique channel that contains an atypical kinase domain at its C-terminus [31,32]. Inhibition of TRPM7 affects the pluripotency and self-renewal of mouse ESCs [33]. They suggested that this effect is caused by decreased phosphorylation of mTOR and subsequent activation of ERK by TRPM7 inhibition. Furthermore, the lack of TRPM7 kinase domain leads to proliferation arrest in mouse ESCs [9]. They suggested that these effects are caused by the disruption of magnesium homeostasis because supplementation of Mg^2+^ rescues the proliferation arrest of those cells. Therefore, TRPM7 is crucial for the maintenance of pluripotency in mouse PSCs and may also play a role in human PSCs.

### 7.3. The Role of TRPV Channels in Pluripotent Stem Cells

TRPV1 KO cell line in human ESCs exhibits normal pluripotency and karyotype, showing this channel is not essential for human PSCs [34]. However, Matsuura et al. reported that TRPV1 activation via heat shock and treatment with its agonist eliminates undifferentiated human iPSCs in differentiated cardiac cell sheet tissues [35]. This is based on the difference in tolerance to TRPV1 activation between iPSCs and differentiated cardiac cells. This result suggests that excess activation of TRPV1 causes a deleterious effect on PSCs. Therefore, moderate activity of this channel is required for the maintenance of those cells. In mouse ESCs, TRPV3 is reported to be present in the endoplasmic reticulum [36]. They showed that TRPV3 activation leads to decreased proliferation via G2/M arrest, showing that TRPV3 is required for self-renewal capacity in mouse ESCs.

## 8. Piezo Channels

Piezo-type mechanosensitive channels are mechanically activated ion channels and regulate cellular mechanotransduction [37]. Two subtypes of the Piezo channel are encoded by *PIEZO1* and *PIEZO2*, and these are important in somatosensation, red blood cell volume regulation and vascular physiology. In PSCs, the expression of these channels and the current via them are shown to be present [38].

### The Role of Piezo Channels in Pluripotent Stem Cells

It was found that mechanosensitive current exists in mouse ESCs, and *Piezo1* expression is detected by transcriptome analysis [38]. Furthermore, KO of Piezo1 in mouse ESCs significantly reduces the proliferation rate without any changes in pluripotency markers [38].

## 9. Cyclic Nucleotide-Gated and Hyperpolarization-Activated Channels

CNG, cyclic nucleotide-gated channels are ion channels that are activated by the binding of cGMP or cAMP [39]. CNG channels are formed by four subunits, either of type A (CNG channel α-1 to 4 encoded by *CNGA1*, *CNGA2*, *CNGA3* and *CNGA4*) or type B (CNG channel β-1 and 3 encoded by *CNGB1* and *CNGB3*), and are crucial in many physiological processes such as vision and pacemaking in the heart [40,41]. HCN channels consist of four members: brain cyclic nucleotide-gated channel (BCNG)-1 and -2 and hyperpolarization-activated cyclic nucleotide-gated potassium channels (HCN)-3 and -4, encoded by *HCN1*, *HCN2*, *HCN3* and *HCN4*, respectively [42]. HCN channels are activated by membrane hyperpolarization, and Na^+^ and K^+^ are permeabilized into cells. Activation of this channel is facilitated by direct interaction with cyclic nucleotides. Among them, only HCN3 is found to be expressed on a protein level in mouse ESCs [43].

### The Role of HCN Channels in Pluripotent Stem Cells

Hyperpolarization-activated inward currents are present in mouse ESCs, and the application of HCN channel blockers, cesium or ZD7288 reduces the proliferation rate via changes in the cell cycle, suggesting that cell cycle progression and proliferation capacity of mouse ESCs could be regulated by HCN channels [43]. Omelyanenko et al. also reported that HCN channel blocker ZD7288 reduces the proliferation rate of mouse ESCs while pluripotency marker expressions are maintained [44].

## 10. Acid-Sensing Ion Channels

Acid-sensing ion channels (ASICs) are mainly proton-gated cation channels that sense pH changes and are activated by acidic pH and nonproton ligands [45]. ASICs are predominantly expressed in the central nervous system and have a role in synaptic functions. There are few reports about their roles in PSCs. However, mRNA of *ASIC2* and *ASIC5* could be expressed in H1 human ESCs (Figure 2A).

## 11. Chloride Channels

There are five classes of chloride channels: cystic fibrosis transmembrane conductance regulator (CFTR), which is activated by cyclic AMP-dependent phosphorylation; calcium-activated chloride channels (CLCAs); voltage-gated chloride channels (CLCNs); ligand-gated chloride channels; and volume-regulated chloride channels [46]. These channels are involved in key cellular events, including cell volume regulation, transepithelial fluid transport, muscle contraction and neuroexcitation [46]. There are few reports about their roles in PSCs. However, mRNA of *CLIC4* and *CLIC5* could be expressed in human ESCs, and their levels seem to be downregulated during differentiation (Figure 2C).

## 12. Transmembrane Channel-like Channels

Transmembrane channel-like proteins 1 (TMC1) and 2 (TMC2) are mechanosensitive ion channels which have a specific function in mechanosensory transduction machinery in hair cells of the inner ear [47]. There are few reports about their roles in PSCs.

## 13. Two-Pore Channels

Two-pore channels (TPCs) are Ca^2+^-permeable endo-lysosomal ion channels, which mediate Ca^2+^ release from acidic organelles in response to cues such as the second messenger, nicotinic acid adenine dinucleotide phosphate (NAADP) [48]. TPCs regulate endosomal function. Therefore, they have many functions in cellular processes. They especially play a role in various infectious diseases [49].

### The Role of TPC Channels in Pluripotent Stem Cells

*TPC2* expression is detected in mouse ESCs by RT-PCR, and this expression decreases during neural differentiation [50]. *TPC2* knockdown promotes mouse ESC differentiation into neural progenitors but inhibits the conversion from neural progenitors to neurons [50]. This report suggests that TPC2 maintains a pluripotent state in PSCs from differentiation cues for neural fate.

## 14. Summary of Channels Expressed in PSCs

In PSCs, the major ion channels of sodium, potassium and calcium are confirmed to be expressed, especially *SCN8A*, *KCNC4*, *KCNK1*, *KCNK5*, *KCNK6*, *KCNQ2*, *KCNS3*, *KCNMB4*, *KCNN2* and *KCNH6* by previous research reports and *SCN4A*, *SCN5A*, *SCN8A*, *KCNE3*, *KCNH1*, *KCNH2*, *KCNH3*, *KCNH6*, *KCNJ10*, *KCNJ11*, *KCNK2*, *KCNK4*, *KCNK5*, *KCNK6*, *KCNK7*, *KCNK12*, *KCNK17*, *KCNMA1*, *KCNQ1*, *KCNQ2*, *KCNS1* and *KCNS3* by microarray analysis compared with EBs. Previous research suggests that potassium channels are more important for the unique properties of PSCs than sodium channels.

Among other channels, the expressions of *TRPC3*, *TRPM7*, *TRPV1* and *PIEZO1* in PSCs have been confirmed by previous reports and their roles in proliferation and/or pluripotency are shown using human or mouse PSCs. Also, the expression and function of HCN channels in mouse ESCs are shown by previous reports. HCN channels could have a role in the proliferation capacity of PSCs. Furthermore, *TPC2* is confirmed to be expressed in mouse ESCs, and its inhibition promotes neural differentiation, showing that the TPC2 channel could have a role in maintaining pluripotency in PSCs. These functional roles of ion channels revealed by previous reports are summarized in Table 1.

## 15. Candidates of Channels Which Have Some Roles in PSCs

The involvement of other channels encoded by *CLIC4*, *CLIC5*, *TMC8*, *TRPM4*, *TRPML2* and *TRPV3*, which are detected by composed data from microarray analysis, is still unknown. As mentioned above, TRPM4 KO human ESCs and iPSCs are normally proliferative and have intact pluripotency [29], showing that *TRPM4* is expressed but does not have critical functions in human PSCs.

CLIC is a chloride intracellular channel. There are six members in the CLIC family (CLIC1, CLIC2, CLIC3, CLIC4, CLIC5 and CLIC6) [51]. So far, the CLIC4 protein (253 amino acids) is the well-studied family member. It is ubiquitously expressed and has been reported to localize to various subcellular compartments, including organelles, plasma membranes, vesicles and centrosomes [52]. CLICs have been extensively studied in cancer and tumor growth and display differential expression and localization in cancer cells during metastasis [53]. Recently, Sanchez et al. have shown that *CLIC4* expression is higher in breast cancers from younger women and those with early-stage metastatic disease [54]. At the cellular level, CLIC4 participates in enhancing TGF-β activity by preventing dephosphorylation of phosphor-SMADs in the nucleus [55]. TGF-β signaling is known to be required to maintain the pluripotent state of human ESCs [56]. In these cells, the TGF-β/SMAD pathway interacts with the master transcription factors, OCT4 and NANOG, which regulate the pluripotent state and differentiation of human ESCs. Then, both OCT4 and NANOG can form a protein complex with SMAD2, and likely SMAD3 (SMAD2/3) [3,57], and these transcription factors tend to co-occupy the genome with OCT4, NANOG and SOX2 in human ESCs [58,59]. Thus, it seems that CLIC4 has some roles in their pluripotency via activation of TGF-β.

*TRPML2* encodes Mucolipin-2, also known as *MCOLN2*, and it is a member of the transient receptor potential mucolipin (TRPML) family consists of three members: TRPML1 (Mucolipin-1), TRPML2 (Mucolipin-2) and TRPML3 (Mucolipin-3) [60]. They are intracellular ion channels of the main Ca^2+^-permeable channels in endo-lysosomal membranes [61]. TRPMLs contain a six transmembrane domain with cytosolic amino- and carboxyl termini, as well as a channel pore located between transmembrane domains (TM) 5 and TM6 [62]. These channels are known to play a role in endosome–lysosome fusion, scission of endo-lysosomal hybrid organelles, autophagy, vacuolar pH regulation, exocytosis and metal homeostasis [62]. Recent studies have shown that Mucolipin-2 may play an important role in the immune system, while some reports suggest an implication of this channel in cancer progression. *TRPML2* expression is elevated in prostate cancer tissues and associated with poor prognosis [63]. Then, overexpression of Mucolipin-2 promoted the proliferation, migration and invasion of prostate cancer cells by regulating the interleukin-1β (IL-1β)/NF-κB pathway. There are some reports about PSCs on the role of NF-κB in maintaining their pluripotency. Undifferentiated human iPSCs show an augmentation of NF-κB activity compared to spontaneously differentiated cells [64]. Inhibition of NF-κB signaling reduces the expression of *OCT4* and *NANOG* and upregulates the differentiation marker *WT1* and *PAX2*. Also, Armstrong et al. showed that inhibition of NF-κB reduces the expression of *OCT4*, *NANOG* and the cell surface marker SSEA4 and induces differentiation of human ESCs [65]. These reports suggest a possibility that Mucolipin-2 has some roles in maintaining the pluripotency of human PSCs via regulating the NF-κB pathway. Future studies are required to elucidate this hypothesis.

*TMC8* encodes a member of transmembrane channel-like proteins. TMC1 and TMC2, the main members of TMC, are known to be pore-forming subunits of mechanosensitive ion channels [66], while the function of TMC8 protein is not well understood. Recently, TMC proteins have been thought to be critical in the carcinogenesis, proliferation and cell cycle of human cancers [67]. The relationship between the function of TMC8 and human cancers has not been studied. However, dysregulation of TMC8 could affect cancer progression, suggesting that this protein regulates the cell cycle in infinitely proliferative cells like PSCs.

## 16. Conclusions

Basically, the representative role of ion channels is signal transduction in response to extracellular stimulation and conduction of excitation in excitable cells like neurons, muscles and cardiac cells. In addition to such a basic function, some of these channel proteins regulate proliferative capacity in infinitely progressive cells like cancer cells and PSCs via modulating the cell cycle. Furthermore, maintaining pluripotency or exit to a differentiated state could be regulated by a few channel proteins via affecting pluripotency factor expression. In this review, other candidates that could regulate such properties of PSCs are proposed based on data from previous reports. It could be important to elucidate their involvement in the unique properties of PSCs to fully understand their physiology.

## Figures and Tables

**Figure 1 biology-13-01043-f001:**
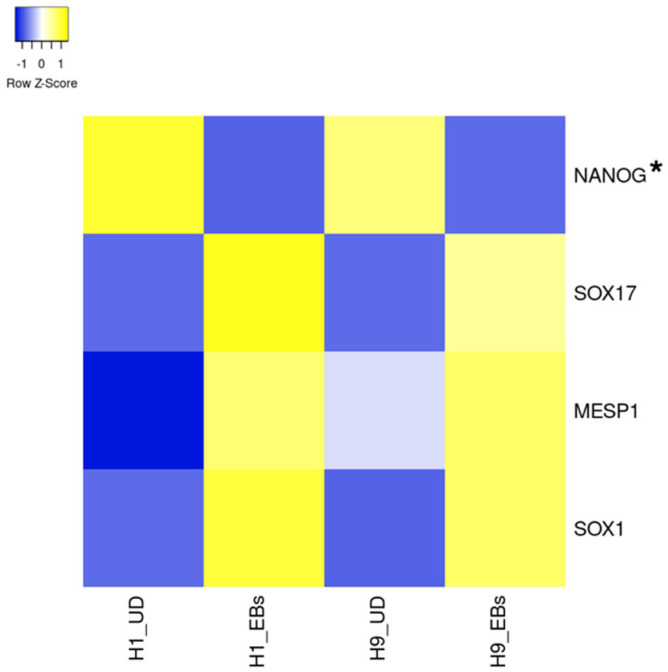
Heatmap of expression levels of pluripotency and differentiation markers in undifferentiated and differentiated human ESCs. The data of mRNA levels in undifferentiated H1 and H9 human ESCs and their differentiated EBs were obtained from the dataset GDS5408 of the NCBI website. Then, a heatmap of pluripotency and differentiation marker levels was created using the Heatmapper website accessed on 7 December 2024 (http://www.heatmapper.ca) [13]. The blue shows downregulation, and the yellow shows upregulation. The asterisk refers to a downregulated gene with a ratio of differentiated EBs to undifferentiated ESCs of less than 0.8-fold in both human ESC lines. UD, undifferentiated.

**Figure 2 biology-13-01043-f002:**
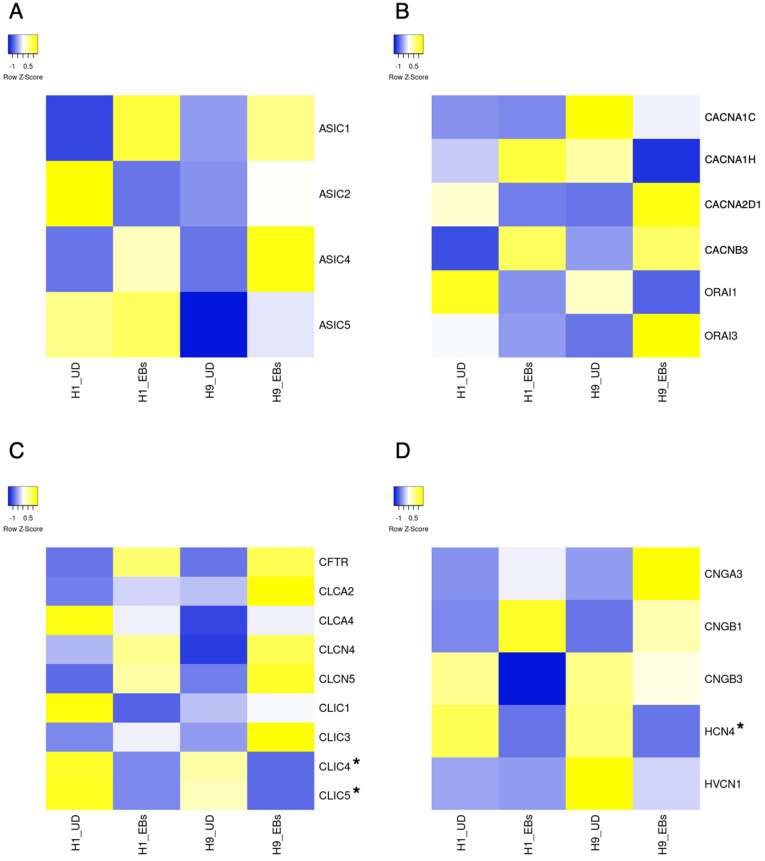
Heatmap of expression levels of ion channel genes in undifferentiated and differentiated human ESCs. The data of mRNA levels in undifferentiated H1 and H9 human ESCs and their differentiated EBs were obtained from the dataset GDS5408 of the NCBI website. Then, heatmaps of *ASICs* (**A**), *CACNAs*, *CACNB*, *ORAIs* (**B**), *CFTR*, *CLCAs*, *CLCNs*, *CLICs* (**C**), *CNGA*, *CNGBs*, *HCN* and *HVCN* (**D**) gene levels were created using the Heatmapper website accessed on 7 December 2024 (http://www.heatmapper.ca) [13]. The blue shows downregulation, and the yellow shows upregulation. Asterisks refer to downregulated genes with a ratio of differentiated EBs to undifferentiated ESCs of less than 0.8-fold in both human ESC lines. UD, undifferentiated.

**Figure 3 biology-13-01043-f003:**
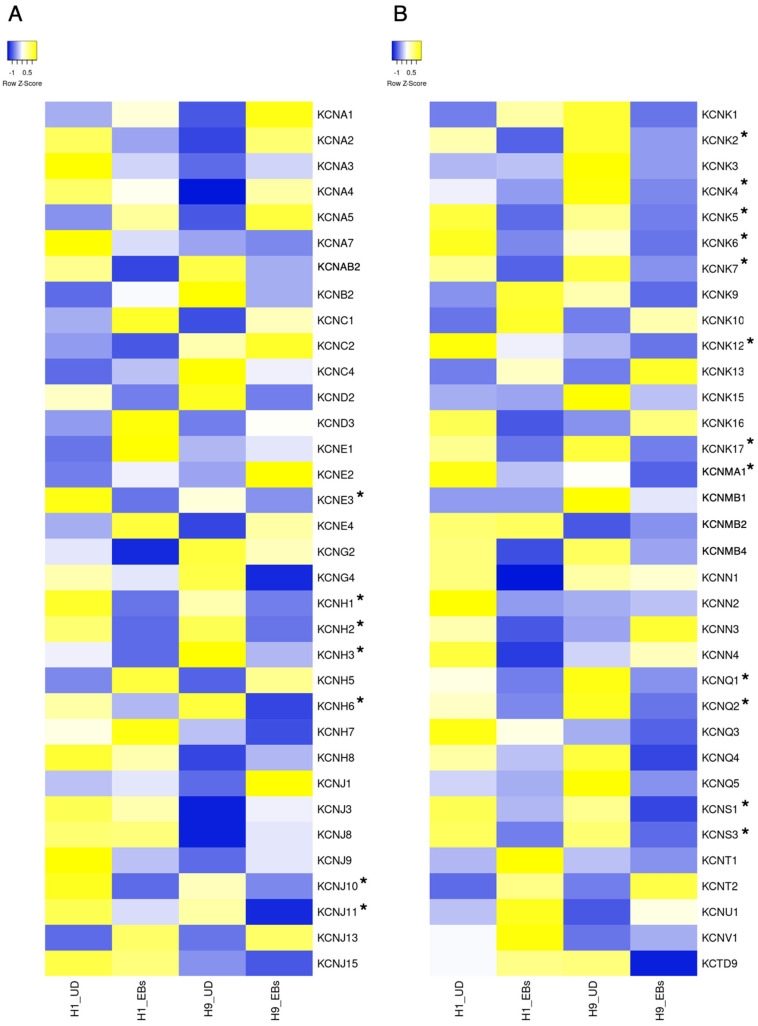
Heatmap of expression levels of ion channel genes in undifferentiated and differentiated human ESCs. The data of mRNA levels in undifferentiated H1 and H9 human ESCs and their differentiated EBs were obtained from the dataset GDS5408 of the NCBI website. Then, heatmaps of *KCNAs*, *KCNBs*, *KCNCs*, *KCNDs*, *KCNEs*, *KCNGs*, *KCNHs*, *KCNJs* (**A**), *KCNKs*, *KCNMs*, *KCNNs*, *KCNQs*, *KCNSs*, *KCNTs*, *KCNU*, *KCNV* and *KCTD* (**B**) gene levels were created using the Heatmapper website accessed on 7 December 2024 (http://www.heatmapper.ca) [13]. The blue shows downregulation, and the yellow shows upregulation. Asterisks refer to downregulated genes with a ratio of differentiated EBs to undifferentiated ESCs of less than 0.8-fold in both human ESC lines. UD, undifferentiated.

**Figure 4 biology-13-01043-f004:**
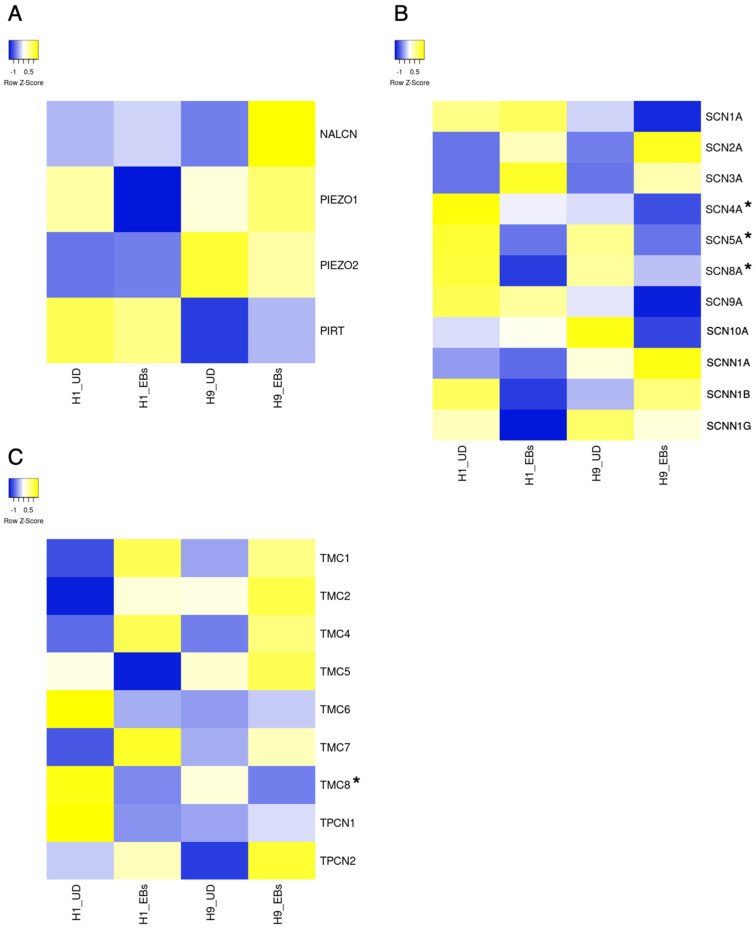
Heatmap of expression levels of ion channel genes in undifferentiated and differentiated human ESCs. The data of mRNA levels in undifferentiated H1 and H9 human ESCs and their differentiated EBs were obtained from the dataset GDS5408 of the NCBI website. Then, heatmaps of *NALCN*, *PIEZOs*, *PIRT* (**A**), *SCNs*, *SCNNs* (**B**) *TMCs* and *TPCNs* (**C**) gene levels were created using the Heatmapper website accessed on 7 December 2024 (http://www.heatmapper.ca) [13]. The blue shows downregulation, and the yellow shows upregulation. Asterisks refer to downregulated genes with a ratio of differentiated EBs to undifferentiated ESCs of less than 0.8-fold in both human ESC lines. UD, undifferentiated.

**Figure 5 biology-13-01043-f005:**
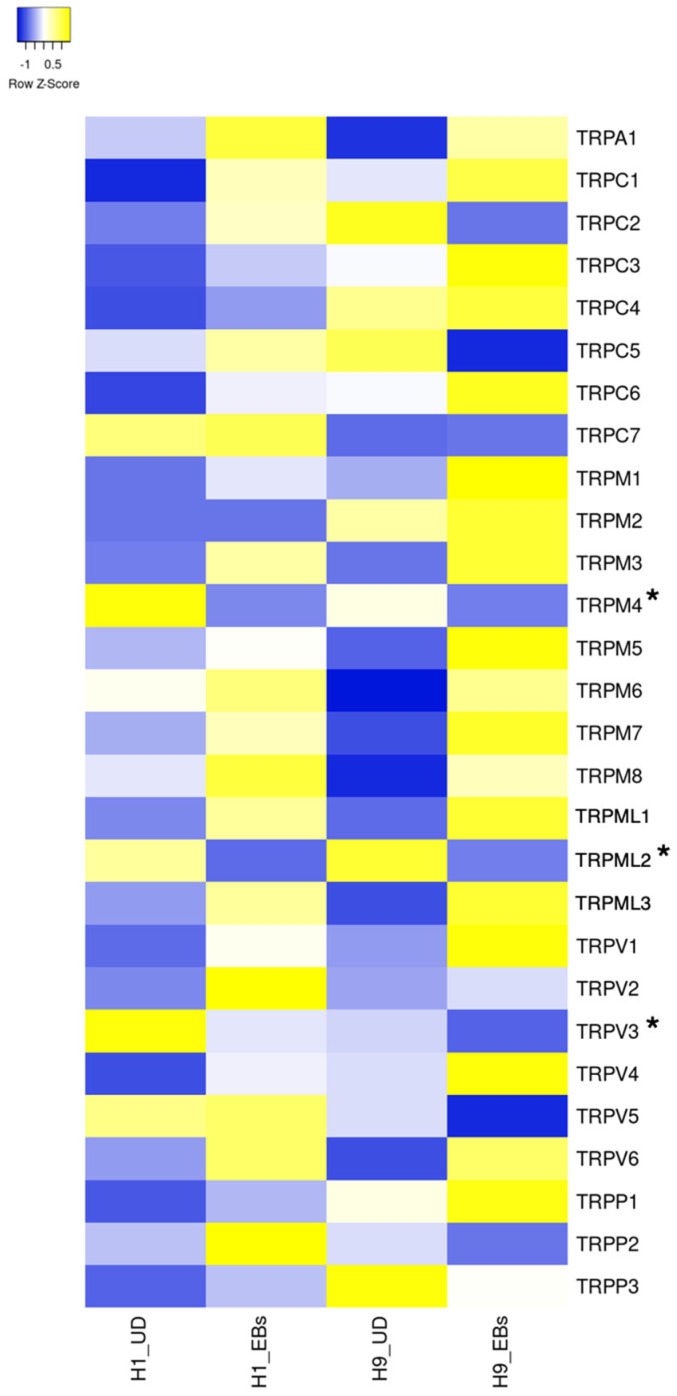
Heatmap of expression levels of ion channel genes in undifferentiated and differentiated human ESCs. The data of mRNA levels in undifferentiated H1 and H9 human ESCs and their differentiated EBs were obtained from the dataset GDS5408 of the NCBI website. Then, heatmaps of *TRPA*, *TRPCs*, *TRPMs*, *TRPMLs*, *TRPVs* and *TRPPs* gene levels were created using the Heatmapper website accessed on 7 December 2024 (http://www.heatmapper.ca) [13]. The blue shows downregulation, and the yellow shows upregulation. Asterisks refer to downregulated genes with a ratio of differentiated EBs to undifferentiated ESCs of less than 0.8-fold in both human ESC lines. UD, undifferentiated.

**Table 1 biology-13-01043-t001:** The summary of ion channels and their known functional roles in pluripotent stem cells.

Ion Channels	Gene	Cell Type	Involved Signaling	Functional Role	Refs.
K^+^ channels	*KCNs*	Human ESCs and iPSCs	Cell cycle	Proliferation	[15,17]
Kv11.2	*KCNH6*	Human ESCs	mTOR pathway	Differentiation	[20]
TRPC3	*Trpc3*	Mouse ESCs	Ca^2+^ signaling	Pluripotency and survival	[28]
TRPM7	*Trpm7*	Mouse ESCs	mTOR and ERK pathway	Pluripotency and self-renewal	[33]
TRPM7	*Trpm7*	Mouse ESCs	Mg^2+^ homeostasis	Proliferation	[9]
Piezo1	*Piezo1*	Mouse ESCs	Mechanotransduction	Proliferation	[38]
HCN channels	*HCNs*	Mouse ESCs	Cell cycle	Proliferation	[43,44]
TPC2	*Tpc2*	Mouse ESCs	Ca^2+^ signaling	Pluripotency	[50]

## Data Availability

The data are available from the corresponding author.

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
