# Peer review of "The Unique Roles of Ion Channels in Pluripotent Stem Cells in Response to Biological Stimuli"

_biology, 2024, doi:10.3390/biology13121043_

Round 1
Reviewer 1 Report
Comments and Suggestions for Authors
General Comments:
The manuscript discussed ion channels and their response to environmental stimuli in PSCs, with compared their expressions and roles between PSCs and their differentiated embryoid bodies, with contributions to pluripotency and differentiation. The study's findings remain valuable, contributing to the understanding of the role of ion channels in PCScs.
Minor Comments:
- Section 2 (page 2). Ion channels: Saying that ion channels which mean a protein that permeates ions to inside the cell this is not true, some channels are involved in releasing ions, e.g. potassium, outside the cell. I would recommend rewriting these sentences like:
Ion channels are proteins that facilitate the movement of ions across cell membranes, playing a crucial role in cellular functions. These channels are found in both the plasma membrane and intracellular membranes and are expressed in organisms ranging from bacteria to humans, where they are essential for life.
- Correct the notation of ions to include superscripts for charge (e.g., Na⁺, K⁺, Ca²⁺, Mg²⁺, Cl⁻) where necessary.
- Gene and protein names are not consistently italicized or capitalized according to standard conventions. Follow accepted nomenclature for genes and proteins. Gene symbols should be capitalized and italicized, while protein symbols should be in regular font. Please correct the whole text (especially sections 5., 6., 14 ) and in Figures.
- When the gene is mentioned I would advise adding the names of proteins, when first mentioned. Please standardize, sometimes protein names are introduced and genes, and sometimes only genes are mentioned, and e.g. in the case of potassium channels the names of genes are different from the names of encoding proteins. Please clearly indicate the protein name for each gene mentioned, although at the first mention in the text.
Author Response
Response to reviewer 1 comments
General Comments:
The manuscript discussed ion channels and their response to environmental stimuli in PSCs, with compared their expressions and roles between PSCs and their differentiated embryoid bodies, with contributions to pluripotency and differentiation. The study's findings remain valuable, contributing to the understanding of the role of ion channels in PCSs.
I appreciate the reviewer’s comments. I have carefully revised the manuscript thoroughly according to the comments
Minor Comments:
- Section 2 (page 2). Ion channels: Saying that ion channels which mean a protein that permeates ions to inside the cell this is not true, some channels are involved in releasing ions, e.g. potassium, outside the cell. I would recommend rewriting these sentences like:
Ion channels are proteins that facilitate the movement of ions across cell membranes, playing a crucial role in cellular functions. These channels are found in both the plasma membrane and intracellular membranes and are expressed in organisms ranging from bacteria to humans, where they are essential for life.
I really appreciate the reviewer’s kind suggestion. Certainly, ion channels release ions to outside the cells. I have changed this sentence to the recommended sentences in this revised manuscript.
- Correct the notation of ions to include superscripts for charge (e.g., Na⁺, K⁺, Ca²⁺, Mg²⁺, Cl⁻) where necessary.
As the reviewer pointed out, I have corrected the notation of ions.
- Gene and protein names are not consistently italicized or capitalized according to standard conventions. Follow accepted nomenclature for genes and proteins. Gene symbols should be capitalized and italicized, while protein symbols should be in regular font. Please correct the whole text (especially sections 5., 6., 14 ) and in Figures.
I have corrected notation of genes and proteins throughout the manuscript.
- When the gene is mentioned I would advise adding the names of proteins, when first mentioned. Please standardize, sometimes protein names are introduced and genes, and sometimes only genes are mentioned, and e.g. in the case of potassium channels the names of genes are different from the names of encoding proteins. Please clearly indicate the protein name for each gene mentioned, although at the first mention in the text.
I have added protein name beside its gene name about all channels at least at the first mention.
Reviewer 2 Report
Comments and Suggestions for Authors
1. Title.
Maybe it would be better to substitute the term “environmental” to “biological”?
2. Chapter 2.
2.1. “Ion channels which mean a protein that permeates ions to inside the cell…”.
This statement is incorrect as ion channels also permeate the ions from the cytosol of the cell to the extracellular milieu (i.e., K channels).
2.2. “Basically, Na and K ions regulate membrane potential inside and out-side the cell“.
What is membrane potential (MP) “inside the cell”? Does author mean the regulation of membrane potential BETWEEN inside and outside of the cells? If it is true, it is better to state that Na and K ions regulate the MP of the plasma membrane
3. Heatmaps of channel expression (Figures).
If I have correctly understood, the * sign means the decrease of specific ion channel expression in differentiated cells, comparing to un-differentiated. If so, the Author should carefully check the correctness of “*” that highlights the decrease of channel expression in differentiated EB. Particularly, the * sign should also be nearby:
Figure2 – CACNA1C, HVCN1
Figure3 - KCNA7, KCNK12, KCNK15, KCNMA1, KCNN1, KCNQ5
Figure4 – SCN4A, SCN9A, TRPV3.
4. Chapter 4.1. Sodium channels
“Using patch-clamp analysis with channel blockers, delayed rectifier K+ currents were observed” – These results should be mentioned in the chapter on K channels
5. Chapter 5. Potassium channels
The Author wrote: “Potassium channels mainly function in ex-citable cells like neurons and cardiomyocyte and its dysfunction cause psychiatric disorder and cardiovascular diseases in humans”
Comment: The role of potassium channels is definitely not restricted to cardiovascular diseases and psychiatric disorders. It is better to refer the readers to the specific up-to-date reviews on K channels in the cells (This could be done similar to that in Sodium channel part, where there is the reference to the specific review [15]).
6. Chapter 6.1.
The data on expression of Orai channels in un-differentiated and differentiated ESCs should be extracted from datasets and plotted as a Heatmap.
7. Chapter 7.
The Author wrote: “TRP channel family is composed from TRPA, TRPC, TRPM, TRPV encoded by TRPA1, TRPC3, TRPC6, TRPM1, TRPM2, TRPM3, TRPM4, TRPM5, TRPM6, TRPM7, TRPM8, TRPV1, TRPV2, TRPV3, TRPV4 and TRPV6”
First concern is that among TRP channel families TRPML and TRPP are missing. The data on their expression in un-differentiated and differentiated ESCs should be extracted from datasets and plotted as a Heatmap.
Second – several members inside specific TRP channel families are also missing (i.e. TRPC1, 2, 5, TRPV5, etc). It is uncertain why the Author had not listed them. The Author should carefully check the missing ion channels from TRP channel family and add them to the Figures.
8. Chapter 7-3. A role of TRPV channels in pluripotent stem cells
The Author state: “TRPV1 KO cell line in human ESCs exhibits normal pluripotency and karyotype, showing this channel is not essential for human PSCs (31). However, Matsuura et al. re-ported that TRPV1 activation via heat shock and treatment with its agonist eliminates un-differentiated human iPSCs in differentiated cardiac cell sheet tissues (32)”
Comment: The expression levels of TRPV1 channels are absent on Figure 4D, whereas its role in iPSCs is stated here. Thus, expression levels of TRPV1 should be added to Heatmaps
9. Summary of channels expressed in PSCs
“In PSCs, the basic channels of sodium, potassium and calcium are confirmed to be expressed”
What means “basic channels”? What mean "In another channels"? Maybe “Among other channels”?
10. Chapter: 16. Summary and outlook
It seems from the current review that there is very limited number of evidences of the roles of various ion channels in regulation of specifically PSCs. It would be better to summarize the channels and their FUNCTIONAL roles (not just evidences of mRNA expression, that was already summarized in heatmaps) as the table. In this table the specific ion channel, cell type and the channel’s impact on cell signalling could be listed.
Author Response
Response to reviewer 2 comments
I appreciate the reviewer’s valuable comments.
- Title.
Maybe it would be better to substitute the term “environmental” to “biological”?
I have changed the title according to the comment.
- Chapter 2.
2.1. “Ion channels which mean a protein that permeates ions to inside the cell…”.
This statement is incorrect as ion channels also permeate the ions from the cytosol of the cell to the extracellular milieu (i.e., K channels).
I appreciate the reviewer’s comment. Certainly, there are channels which permeate ions from inside to outside of the cell. I have changed this sentence to “Ion channels are proteins that facilitate the movement of ions across cell membranes, playing a crucial role in cellular functions. These channels are found in both the plasma membrane and intracellular membranes and are expressed in organisms ranging from bacteria to humans, where they are essential for life.” In this revised manuscript.
2.2. “Basically, Na and K ions regulate membrane potential inside and out-side the cell“.
What is membrane potential (MP) “inside the cell”? Does author mean the regulation of membrane potential BETWEEN inside and outside of the cells? If it is true, it is better to state that Na and K ions regulate the MP of the plasma membrane
I made a mistake in text representation. I have corrected this sentence as the reviewer suggested.
- Heatmaps of channel expression (Figures).
If I have correctly understood, the * sign means the decrease of specific ion channel expression in differentiated cells, comparing to un-differentiated. If so, the Author should carefully check the correctness of “*” that highlights the decrease of channel expression in differentiated EB. Particularly, the * sign should also be nearby:
Figure2 – CACNA1C, HVCN1
Figure3 - KCNA7, KCNK12, KCNK15, KCNMA1, KCNN1, KCNQ5
Figure4 – SCN4A, SCN9A, TRPV3.
As the reviewer commented, * means decreased genes in differentiated cells. And I apologize for missing some genes which actually decrease in differentiated cells, but were not marked with *. In this revised manuscript, to improve the definition of the decreased genes in differentiated EBs, I have changed the mean of * to the genes which decrease less than 0.8-fold in the ratio of differentiated EBs to undifferentiated ESCs. By this change, the genes marked with * are renewed as blow;
Figure 1 – NANOG.
Figure 2 - CLIC4, CLIC5, HCN4.
Figure 3 – KCNE3, KCNH1, KCNH2, KCNH3, KCNH6, KCNJ10, KCNJ11
KCNK2, KCNK4, KCNK5, KCNK6, KCNK7, KCNK12, KCNK17, KCNMA1, KCNQ1, KCNQ2, KCNS1, KCNS3.
Figure 4 – SCN4A, SCN5A, SCN8A, TMC8.
Figure 5 – TRPM4, TRPML2 (MCOLN2), TRPV3
Accompanied by this change, I have revised the text throughout the manuscript.
- Chapter 4.1. Sodium channels
“Using patch-clamp analysis with channel blockers, delayed rectifier K+ currents were observed” – These results should be mentioned in the chapter on K channels
I have removed this sentence in chapter 4.1. Instead, the sentence “Non-selective potassium channel blocker…” have been written in chapter 5.1 of this revised manuscript.
- Chapter 5. Potassium channels
The Author wrote: “Potassium channels mainly function in excitable cells like neurons and cardiomyocyte and its dysfunction cause psychiatric disorder and cardiovascular diseases in humans”
Comment: The role of potassium channels is definitely not restricted to cardiovascular diseases and psychiatric disorders. It is better to refer the readers to the specific up-to-date reviews on K channels in the cells (This could be done similar to that in Sodium channel part, where there is the reference to the specific review [15]).
I have added two review articles as reference [18,19] to the chapter 5.
- Chapter 6.1.
The data on expression of Orai channels in un-differentiated and differentiated ESCs should be extracted from datasets and plotted as a Heatmap.
I have added the expression data of ORAI1 and ORAI3 to Figure 2B.
- Chapter 7.
The Author wrote: “TRP channel family is composed from TRPA, TRPC, TRPM, TRPV encoded by TRPA1, TRPC3, TRPC6, TRPM1, TRPM2, TRPM3, TRPM4, TRPM5, TRPM6, TRPM7, TRPM8, TRPV1, TRPV2, TRPV3, TRPV4 and TRPV6”
First concern is that among TRP channel families TRPML and TRPP are missing. The data on their expression in un-differentiated and differentiated ESCs should be extracted from datasets and plotted as a Heatmap.
As reviewer pointed out, I have missed to include TRPMLs and TRPPs in the TRP channel family. I have added them to the text in chapter 7 and the data of their expression levels to Figure 5 of this revised manuscript.
Second – several members inside specific TRP channel families are also missing (i.e. TRPC1, 2, 5, TRPV5, etc). It is uncertain why the Author had not listed them. The Author should carefully check the missing ion channels from TRP channel family and add them to the Figures.
I apologize for my oversight of dataset. I have added the expression data of TRPC1, TRPC2, TRPC4, TRPC5, TRPC7, TRPML1, TRPML2, TRPML3, TRPV1, TRPV5, TRPP1, TRPP2 and TRPP3 to Figure 5 of this revised manuscript. Alternative name of TRPML2 is MCOLN2, therefore, the data of MCOLN2 in Figure 4A has been moved to Figure 5.
- Chapter 7-3. A role of TRPV channels in pluripotent stem cells
The Author state: “TRPV1 KO cell line in human ESCs exhibits normal pluripotency and karyotype, showing this channel is not essential for human PSCs (31). However, Matsuura et al. re-ported that TRPV1 activation via heat shock and treatment with its agonist eliminates un-differentiated human iPSCs in differentiated cardiac cell sheet tissues (32)”
Comment: The expression levels of TRPV1 channels are absent on Figure 4D, whereas its role in iPSCs is stated here. Thus, expression levels of TRPV1 should be added to Heatmaps
I have added the expression data of TRPV1 to Figure 5 of this revised manuscript.
- Summary of channels expressed in PSCs
“In PSCs, the basic channels of sodium, potassium and calcium are confirmed to be expressed”
What means “basic channels”? What mean "In another channels"? Maybe “Among other channels”?
I have revised the words “basic channels” to “major ion channels” and “In another channels” to “Among other channels” as the reviewer suggested.
- Chapter: 16. Summary and outlook
It seems from the current review that there is very limited number of evidences of the roles of various ion channels in regulation of specifically PSCs. It would be better to summarize the channels and their FUNCTIONAL roles (not just evidences of mRNA expression, that was already summarized in heatmaps) as the table. In this table the specific ion channel, cell type and the channel’s impact on cell signaling could be listed.
I appreciate this suggestion. I have prepared and added a table summarizing ion channels and their roles in pluripotent stem cells.
Round 2
Reviewer 2 Report
Comments and Suggestions for Authors
All my comments were properly addressed
The paper could be accepted for the publication